# Attitudes, Roles, and Competencies of Clinical Psychologists Regarding Euthanasia Due to Unbearable Mental Suffering

**DOI:** 10.3390/ejihpe15110228

**Published:** 2025-11-05

**Authors:** Dennis Demedts, Wouter Drijkoningen, Johan Bilsen

**Affiliations:** Mental Health and Wellbeing Research Group (MENT), Vrije Universiteit Brussel, 1090 Brussels, Belgiumjohan.bilsen@vub.be (J.B.)

**Keywords:** attitudes, euthanasia, professional competence, psychologists, role, unbearable mental suffering

## Abstract

Since the enactment of Belgium’s euthanasia law in 2002, clinical psychologists have played an increasingly significant role in the multidisciplinary approach to euthanasia, particularly regarding cases involving unbearable mental suffering (UMS euthanasia). This cross-sectional study examined the attitudes, roles, and competencies of clinical psychologists in Flanders concerning UMS euthanasia using an online questionnaire (*n* = 242). The survey explored attitudes towards UMS euthanasia, as well as self-perceived competencies and involvement. Results indicate that most psychologists hold a generally positive stance towards UMS euthanasia and believe in supporting patient requests under appropriate conditions. Their roles are seen as essential in decision-making, exploring alternatives, and providing aftercare for families, though opinions vary about their involvement during the actual procedure. Demographic factors such as age, gender, work setting, and prior involvement in euthanasia showed no significant influence on attitudes; however, greater knowledge and skills were reported among those in specialized settings, palliative care, or with prior euthanasia involvement. A pronounced lack of training and education in this area was reported, highlighting the need for targeted measures. The findings underscore the necessity of clarifying psychologists’ roles, expanding legal guidelines, and improving training to enhance care quality in the context of UMS euthanasia.

## 1. Introduction

Life-ending practices represent an increasingly prominent and controversial issue in contemporary healthcare ethics. Across Europe and globally, a growing number of countries have legalized or expanded access to medically assisted dying, sparking substantial social and professional debate ([26]). Two primary forms are distinguished: euthanasia, in which a physician administers a life-ending drug at the patient’s explicit request ([4]), and physician-assisted suicide, where the patient self-administers the prescribed medication ([21]). Together, these practices are commonly referred to as medically assisted dying and are considered only when suffering is deemed unbearable and unrelievable through alternative therapies ([6]).

In Belgium, the Netherlands, Luxembourg, Canada, Spain, New Zealand, and several Australian states, both euthanasia and physician-assisted suicide are legal. In contrast, countries such as Austria, Switzerland, and several U.S. jurisdictions restrict legal access to physician-assisted suicide only ([19]). Within Belgium, euthanasia may also be granted in cases of unbearable mental suffering (UMS) arising from non-terminal or psychiatric disorders considered both untreatable and hopeless ([17]). UMS encompasses a wide spectrum of psychological distress, from severe psychiatric illness to existential despair without an identifiable medical diagnosis. It is a deeply subjective and complex phenomenon, relying primarily on patients’ self-reports and clinical interpretation rather than objective biomarkers. Its evaluation requires careful multidisciplinary deliberation involving both medical and mental health experts. Therefore, under the Belgian Euthanasia Act (2002), specific legal criteria must be satisfied: the patient must be an adult and competent, experiencing a medically hopeless situation resulting in unbearable physical and/or mental suffering caused by an incurable condition or accident, and the euthanasia request must be voluntary, well-considered, and repeated. The attending physician is required to fully inform the patient about their health status, potential therapies, prognosis, and palliative options, and must consult a second, independent physician. In non-terminal or psychiatric cases, a third, specialized physician must be consulted, and a one-month reflection period is mandatory ([3]).

Although substantial research has explored physicians’ and nurses’ attitudes toward euthanasia, traditionally examined due to their direct involvement in physical care, empirical insight into clinical psychologists’ perspectives remains limited ([9]; [28]). Yet, clinical psychologists hold a distinctive position in this context: they assess decisional capacity, evaluate UMS, and contribute to multidisciplinary end-of-life deliberations. Their psychological expertise is crucial to ensuring ethically sound decision-making, but their roles and attitudes have received little systematic investigation. Understanding their views and professional competencies is essential for developing integrative care models that balance ethical sensitivity with psychosocial expertise.

The ethical challenges surrounding UMS euthanasia highlight a central tension in professional psychology: reconciling respect for patient autonomy with the duty to protect vulnerable individuals. Psychologists operate at this intersection, committed to supporting self-determination and alleviating mental suffering while guarding against decisions influenced by impaired judgment or fluctuating mental states. Bioethical and psychological frameworks emphasize the importance of balancing autonomy and protection, recognizing that the subjective nature of suffering complicates assessments of capacity and consent ([2]). Situating the present study within these frameworks underscores the need to clarify clinical psychologists’ roles, ethical orientations, and required competencies regarding UMS euthanasia.

While the euthanasia debate is global, Belgium provides a particularly instructive legal and healthcare context, featuring a mature regulatory framework and established multidisciplinary procedures. Conducting this study in Belgium therefore offers a valuable opportunity to explore psychologists’ professional views within one of the world’s most developed systems of (UMS) euthanasia governance. Findings are expected to enrich international understanding despite contextual differences across countries. Guided by these considerations, the present study addresses the following question: What are the attitudes, roles, and competencies of clinical psychologists regarding UMS euthanasia?

Based on prior research and identified knowledge gaps, the study tests the following hypotheses:Clinical psychologists in Flanders will generally exhibit positive attitudes toward UMS euthanasia, reflecting support for patient self-determination and professional involvement in euthanasia care.Attitudes, perceived roles, and self-reported competencies will vary according to demographic (e.g., age, gender) and professional factors (e.g., healthcare echelon, prior experience with euthanasia requests).Despite positive attitudes, psychologists will report notable gaps in knowledge and skills, highlighting the need for enhanced education and training initiatives.

## 2. Materials and Methods

### 2.1. Research Design

This study employed a cross-sectional quantitative design using an online questionnaire. In this study, UMS is conceptualized in accordance with the criteria established by the Belgian Euthanasia Act (2002) ([3]). UMS refers to a subjective experience of severe psychological distress that the patient perceives as unbearable and is considered incurable and untreatable following comprehensive multidisciplinary evaluation. In contrast to somatic suffering—which can often be substantiated by objective medical findings—the assessment of UMS relies primarily on clinical judgment, patient self-report, and the integration of psychiatric diagnoses and existential dimensions. Within the structured Belgian legal framework, specialized clinicians are required to conduct detailed evaluations of the patient’s mental state, identify underlying conditions, and determine whether the experienced suffering meets the legal and clinical thresholds for UMS euthanasia.

### 2.2. Study Population

The study population consisted of clinical psychologists practicing in Flanders (Dutch-speaking region of Belgium), working across primary, secondary, and tertiary healthcare settings. In Belgium, 14,845 clinical psychologists were employed in 2021, with 8367 in the Flemish community and 6478 in the French-speaking community ([5]).

### 2.3. Participant Selection and Recruitment

Participants in this study were recruited primarily via the Flemish Association of Clinical Psychologists (VVKP), representing over 2500 clinical psychologists in Flanders, as well as the Flemish Association for Psychologists in General Hospitals (VVPAZ). Additional recruitment occurred through psychiatric centers, palliative care support teams, and primary care psychologists in Flanders. To facilitate survey dissemination, email invitations were sent to the abovementioned organizations and channels, requesting they share the online questionnaire broadly among their members. The survey was preceded by an information letter, outlining the study purpose and emphasizing participants’ right to withdraw at any time.

### 2.4. Data Collection and Analysis

Participants first answered demographic questions (age, gender, healthcare echelon, palliative care team membership, specialized psychiatric work, and prior euthanasia involvement) before completing the main survey. The UMS-EAS-NL questionnaire comprised 38 items adapted from prior research on UMS euthanasia attitudes in nursing students ([11], [12]) and derived from the validated Euthanasia Attitude Scale ([27]; [25]). The attitude section included four thematic domains: Ethical considerations, Practical considerations, Treasuring life, and Naturalistic beliefs ([8]). Items were rated on a 5-point Likert scale (1 = strongly disagree to 5 = strongly agree), with negatively worded items reverse-coded. Higher scores indicated greater acceptance of UMS euthanasia.

The scale demonstrated good internal consistency (Cronbach’s alpha = 0.812; McDonald’s omega = 0.838) and construct validity in previous studies ([11]). It also differentiated attitudes by prior euthanasia involvement in the current sample. The second section assessed perceived roles and competencies, developed through literature review and expert consensus with adaptations for clinical psychologists, also rated on a 5-point Likert scale ([12]).

#### Data Analysis

Descriptive statistics (means, standard deviations, frequencies) summarized participant characteristics and survey responses. Attitude analyses involved comparing each thematic domain’s scores against a neutral midpoint (“neither agree nor disagree”) to determine overall positive or negative stances. Group comparisons were conducted using one-way ANOVA tests for demographic variables such as age groups and healthcare echelon, and independent samples *t*-tests to assess differences based on gender, palliative care team membership, specialized psychiatric work, and prior euthanasia involvement. Effect sizes (partial eta squared for ANOVAs and Cohen’s *d* for *t*-tests) were calculated to quantify the magnitude of differences. Statistical significance was accepted at *p* < 0.05. All data were collected anonymously using Qualtrics (Provo, UT, USA) and analyzed with IBM SPSS Statistics version 29.0 (Armonk, NY, USA).

### 2.5. Ethical Considerations

This study received approval from the Medical Ethics Committee of the Universitair Ziekenhuis Brussel and Vrije Universiteit Brussel on 15 May 2024 (EC-2024-017), in accordance with ethical standards for research involving healthcare professionals.

To safeguard privacy, data collected were limited to professional attitudes, roles, and competencies; no personal data capable of identifying participants or information about their patients were gathered. Participation was voluntary, anonymous, and could be discontinued at any time without penalty. No incentives were offered for participation.

Prior to survey completion, participants reviewed an informed consent letter explaining the study’s purpose, responsible parties, and data storage measures. Data were stored securely within Qualtrics (Provo, UT, USA) and SPSS (Armonk, NY, USA), with adherence to Vrije Universiteit Brussel guidelines and ethical research protocols.

## 3. Results

### 3.1. Participant Characteristics

After excluding incomplete responses, the final sample consisted of 242 clinical psychologists practicing in various healthcare settings in Flanders, Belgium. The professional distribution across healthcare echelons was: 76 participants (31.4%) worked in primary care, 89 (36.8%) in secondary care, and 77 (31.8%) in tertiary care. In terms of specialized work environments, 78 respondents (32.2%) were employed in specialized psychiatric institutions, while the majority, 164 (67.8%), worked in other settings. A small proportion, 19 psychologists (7.9%), were active members of a palliative care support team. The gender distribution revealed a clear majority of female psychologists (*n* = 215; 88.8%) compared to male psychologists (*n* = 27; 11.2%). No participants identified as another gender. With regard to age, nearly half of the sample (49.6%, *n* = 120) fell within the 25–34-year age group, followed by 64 (26.4%) aged 35–44 years. The 45–54-year and 55–64-year categories accounted for 12.8% (*n* = 31) and 6.6% (*n* = 16), respectively, while the youngest group, aged 18–24 years, comprised 4.5% (*n* = 11) of the participants. Regarding prior involvement in UMS euthanasia cases: 152 psychologists (62.8%) reported having been involved at some stage of the process, ranging from receiving a request, participating in decision-making, being present during the procedure, to providing aftercare.

### 3.2. Conceptual Model of Attitudes, Roles, and Competencies of Clinical Psychologists Regarding UMS Euthanasia

Figure 1 presents a conceptual framework illustrating the interrelationships between attitudes, roles, and competencies of clinical psychologists in the context of euthanasia for unbearable mental suffering (UMS). The model visualizes how supportive attitudes—such as endorsement of patient autonomy and underlying ethical beliefs—influence psychologists’ perceptions and enactment of their professional roles within multidisciplinary euthanasia care teams. Concurrently, knowledge, practical skills, and training constitute critical competencies that enhance role acceptance and effective participation in the euthanasia process. Ethical tensions, specifically the balance between respecting patient autonomy and protecting vulnerable individuals, mediate the dynamics between attitudes and role perceptions.

### 3.3. Attitudes Towards UMS Euthanasia

The results of the questionnaire assessing attitudes toward UMS euthanasia are presented in detail below (Table 1) and are summarized in Table 2. Overall, results indicated that clinical psychologists hold a predominantly positive attitude toward UMS euthanasia. The mean total score was 79.13, which is statistically significant above the neutral midpoint score of 63 (t(241) = 23.81, *p* < 0.001, Cohen’s *d* = 1.53 [95% confidence interval: 1.34, 1.72]).

#### 3.3.1. Theme 1: Ethical Considerations

For the theme “ethical considerations”, the average score is 43.96, which is significantly higher than the neutral average of 33 (t(241) = 25.72, *p* < 0.001, Cohen’s *d* = 1.65 [95% confidence interval: 1.46, 1.85]).

Within this domain, a large majority (88.0%) supported the right to decide to die in cases of unbearable mental suffering, while 3.4% disagreed. In contrast, 39.8% agreed with the statement that ending life out of compassion is wrong. Most respondents (83.9%) agreed that UMS euthanasia should be accepted in contemporary society. A total of 93.0% rejected the view that UMS euthanasia can never be an appropriate solution. UMS euthanasia was considered by the majority (87.2%) to be able to offer help at the right time and place. Most (79.4%) described it as a humane act, and 92.1% disagreed with the statement that it should not be legally permitted. Responses were divided regarding the statement that euthanasia should be applied when a person with psychiatric illness is considered untreatable: 38.5% agreed, 27.7% were neutral, and 33.9% disagreed. Furthermore, 92.5% disagreed that ending a person’s life is wrong under all circumstances. Similarly, 72.7% considered UMS euthanasia acceptable in situations with no hope of recovery. Finally, 90.9% felt that UMS euthanasia provides an opportunity to die with dignity.

#### 3.3.2. Theme 2: Treasuring Life

For the theme “treasuring life”, the average score is 15.06, which is significantly higher than the neutral average of 12 (t(241) = 15.23, *p* < 0.001, Cohen’s *d* = 0.98 [95% confidence interval: 0.83, 1.13]).

In this domain, 71.9% agreed that the patient’s age should play a role in decisions regarding euthanasia. When considering cases where a psychologically incurable patient is increasingly concerned about being a burden to family, 12.8% supported euthanasia in such circumstances, 47.1% did not support it, and 40.1% were neutral. More than half of participants (58.2%) believed that UMS euthanasia would not lead to misuse. Confidence in the Belgian medical system to carry out UMS euthanasia correctly was expressed by 57.5%.

#### 3.3.3. Theme 3: Practical Considerations

For the theme “practical considerations”, the average score is 13.39, which is significantly higher than the neutral average of 12 (t(241) = 9.88, *p* < 0.001, Cohen’s *d* = 0.64 [95% confidence interval: 0.50, 0.77]).

A large majority (93.4%) disagreed that euthanasia should be performed only in cases of physical suffering and not mental suffering. Furthermore, 56.6% disagreed with the statement that everyone has the duty to protect and maintain life, and 61.6% disagreed with the view that one of the main ethical duties of physicians is to extend life, not to end it.

#### 3.3.4. Theme 4: Naturalistic Beliefs

For theme “naturalistic beliefs”, the average score is 6.72, which is higher than the neutral average of 6 (t(241) = 8.69, *p* < 0.001, Cohen’s *d* = 0.56 [95% confidence interval: 0.42, 0.69]).

Almost half of participants (49.6%) agreed that a person should not be kept alive by machines, while 39.3% were neutral. Responses to the statement that natural death is a remedy for suffering were more evenly split, with 46.3% answering “neither agree nor disagree.”

#### 3.3.5. Attitudes in Relation to Demographics

The independent *t*-test showed no statistically significant difference in attitudes toward UMS euthanasia between male and female participants. This was the case both when equal variances were assumed (t(240) = 0.19, *p* = 0.854, two-tailed) and when equal variances were not assumed (t(30.68) = 0.16, *p* = 0.874, two-tailed). The mean difference between men and women was 0.40, with a standard error of 2.16. The 95% confidence interval for this difference ranged from −3.85 to 4.64, indicating a small and statistically non-significant difference. Effect sizes—Cohen’s *d* (0.04), Hedges’ correction (0.04), and Glass’s delta (0.04)—suggest only a very small and practically negligible difference in attitudes between men and women, with the 95% confidence interval encompassing a minimal effect.

The one-way ANOVA revealed no statistically significant differences in attitudes between age categories (F(4, 237) = 0.47, *p* = 0.760). Effect sizes were also very small: an eta-squared of 0.01 indicated that only 1% of the variance in attitudes could be explained by age group. Negative values for epsilon-squared (−0.01) and omega-squared (−0.01) suggest even smaller effect sizes, likely due to the limited variability among groups. The Welch test, conducted due to unequal sample sizes across age groups, confirmed these results (*p* = 0.675).

The one-way ANOVA showed no statistically significant differences in mean attitude scores toward UMS euthanasia among psychologists working in primary, secondary, and tertiary care settings (F(2, 239) = 1.04, *p* = 0.354). The Welch test confirmed this finding (*p* = 0.368). Effect sizes were minimal: eta-squared was 0.01, indicating that only 1% of the variance in attitudes could be attributed to differences between these groups. Epsilon-squared (0.00) and omega-squared (0.00) similarly suggested negligible variation.

The independent *t*-test indicated no statistically significant difference in attitudes between psychologists working in specialized centers and those not working in such centers. This was consistent whether equal variances were assumed (t(240) = −1.796, *p* = 0.074, two-tailed) or not assumed (t(149.48) = −1.79, *p* = 0.076, two-tailed). The mean difference between the two groups was −2.60 (SE = 1.44), with a 95% CI from −5.43 to 0.25, reflecting a small and non-significant difference. Effect sizes—Cohen’s *d* (−0.25), Hedges’ correction (−0.25), and Glass’s delta (−0.25)—suggest slightly lower average attitude scores among psychologists working in specialized institutions, but the effect size is small and of limited practical significance, as the confidence interval spans from a small negative effect to nearly zero.

The independent *t*-test found no statistically significant difference between psychologists working in a palliative care support team and those who were not. This was consistent whether equal variances were assumed (t(240) = −0.89, *p* = 0.372, two-tailed) or not assumed (t(19.49) = −0.66, *p* = 0.520, two-tailed). The mean difference was −2.25 (SE = 2.52), with a 95% CI from −7.22 to 2.71, indicating a small and statistically non-significant difference. Effect sizes—Cohen’s *d* (−0.21), Hedges’ correction (−0.21), and Glass’s delta (−0.22)—indicate slightly lower scores for those working in palliative care teams, but these effects are minimal and of little practical significance, given the wide confidence interval.

The independent *t*-test indicated no statistically significant difference in attitudes toward UMS euthanasia between psychologists with prior involvement in euthanasia and those without such experience. This finding held for the equal variances assumed test (t(240) = −0.71, *p* = 0.477, two-tailed) and the unequal variances assumed test (t(195.95) = −0.72, *p* = 0.471, two-tailed). The mean difference was −1.00 (SE = 1.40), with a 95% CI from −3.76 to 1.76. Effect sizes were small—Cohen’s *d* (−0.10), Hedges’ correction (−0.09), and Glass’s delta (−0.10)—indicating slightly lower attitude scores among those with prior involvement, though the difference is negligible in practical terms.

The multiple regression analysis with “attitude” as the dependent variable and the variables “age categories”, “gender”, “care setting”, “specialized center”, “palliative care support team” and “prior involvement” as predictors shows that these predictors together hardly explain any variation in attitude (R^2^ = 0.023, Adj. R^2^ = −0.002). None of the predictors were statistically significant (*p* > 0.05).

### 3.4. Role of Clinical Psychologists Regarding UMS Euthanasia

The results of the questionnaire assessing the role of clinical psychologists regarding UMS euthanasia are presented in Table 3.

A large majority of respondents (87.2%) indicated that they play a crucial role in the decision-making process regarding UMS euthanasia requests. Furthermore, 96.3% considered it their duty to work with the patient to explore alternative solutions before a final decision is made. In addition, 88.0% stated that it is their responsibility to ensure that the patient is fully informed about the nature and consequences of euthanasia. Almost all respondents (98.8%) supported a multidisciplinary approach to the euthanasia process. Views were mixed on whether a psychologist is better placed than a physician to assess a euthanasia request (26.8% agreed, 35.6% disagreed, and 37.6% were neutral).

Responses were divided regarding whether it is part of the psychologist’s role to be present during the actual performance of euthanasia. The largest proportion (40.5%) remained neutral, 31.8% agreed with such a role, and 27.7% disagreed. A majority of respondents (71.1%) considered it part of their role to provide support to the patient’s family and close relatives after death. More than three-quarters of respondents (76.5%) agreed that a psychologist with ethical objections to euthanasia should be able to transfer the patient’s care to a colleague.

### 3.5. Competencies of Clinical Psychologists Regarding UMS Euthanasia

The results of the questionnaire assessing the competencies of clinical psychologists regarding UMS euthanasia are presented in Table 4.

Almost half of the respondents (47.1%) indicated that they do not have sufficient knowledge to adequately handle a euthanasia request from a patient.

With respect to skills, 37.6% stated that they are not adequately equipped to deal with a euthanasia request, whereas 45.0% believed they do possess the necessary skills.

The majority of respondents (94.2%) stated that UMS euthanasia had not been sufficiently addressed during their training. There was strong consensus (89.0%) that more attention should be devoted to euthanasia during training. A substantial proportion of respondents (70.2%) supported the introduction of more simulation-based training within psychology education.

#### 3.5.1. Knowledge in Relation to Demographics

The evaluation of the item “I have sufficient knowledge to handle a patient’s euthanasia request” revealed several statistically significant findings in relation to the demographic characteristics of clinical psychologists.

The ANOVA analysis showed no statistically significant differences in self-rated knowledge about euthanasia between age categories of psychologists. Mean scores ranged from 2.68 for those aged 25–34 years to 3.00 for those aged 35–44 years. With an F value of 0.84 and a *p* value of 0.50, there were no statistically significant variations between age groups. The effect size was small (η^2^ = 0.01), indicating that age has a limited influence on self-assessed knowledge about euthanasia.

Psychologists working in different healthcare echelons demonstrated statistically significant differences in their knowledge ratings. Primary care psychologists reported a lower mean score (M = 2.39, SD = 1.18) compared with secondary care psychologists (M = 3.03, SD = 1.20) and tertiary care psychologists (M = 2.99, SD = 1.20). The ANOVA yielded an F value of 7.01 and a *p* value of 0.001. The effect size was moderate (η^2^ = 0.06), suggesting that the work setting has a relatively substantial influence on perceived knowledge about euthanasia. Welch’s test confirmed this finding (*p* = 0.001).

There were clinically significant differences in knowledge ratings between psychologists working in a palliative care support team (M = 3.47, SD = 1.12) and those who were not (M = 2.76, SD = 1.22). The *t*-test produced a t value of 2.46 and a *p* value of 0.007. The effect size, measured by Cohen’s *d*, was 0.59, reflecting a moderate effect.

Psychologists who had prior involvement in euthanasia reported higher knowledge scores (M = 3.26, SD = 1.14) than those without such experience (M = 2.08, SD = 0.97). Levene’s test for equality of variances was significant (*p* = 0.001), indicating unequal variances between groups. Accordingly, Welch’s *t*-test was applied, yielding a t value of 8.50 and a *p* value of < 0.001 for both one- and two-tailed tests. The effect size was large (Cohen’s *d* = 1.09), suggesting that prior euthanasia experience is strongly associated with greater self-reported knowledge.

A multiple linear regression analysis was conducted including demographic and workplace-related variables. Independent variables comprised age categories, gender, care setting, specialized center, palliative care support team and prior involvement. The regression model was statistically significant and explained 25.3% of the variance in “I have sufficient knowledge to handle a patient’s euthanasia request” (R^2^ = 0.253, Adj. R^2^ = 0.234, F(6, 235) = 13.24, *p* < 0.001). Within the model, level of care setting (β = 0.22, *p* = 0.037) and prior involvement (β = −1.07, *p* < 0.001) emerged as significant predictors. Age, gender, palliative care support team, and specialized center did not show significant associations.

#### 3.5.2. Competencies in Relation to Demographics

The evaluation of the item “I have sufficient skills to handle a patient’s euthanasia request” revealed several statistically significant findings in relation to the demographic characteristics of clinical psychologists.

The ANOVA comparing self-rated skills across age categories revealed no statistically significant differences. Mean scores ranged from 3.00 for those aged 25–34 years to 3.48 for those aged 45–54 years. With an F value of 1.80 and a *p* value of 0.129, no significant variations were found. The effect size was small (η^2^ = 0.03), suggesting only a limited influence of age on perceived skills in handling euthanasia requests.

Statistically significant differences emerged in skill ratings between psychologists working in different healthcare echelons. Primary care psychologists reported a lower mean score (M = 2.66, SD = 1.27) than secondary care psychologists (M = 3.24, SD = 1.17) and tertiary care psychologists (M = 3.27, SD = 1.15). The ANOVA yielded an F value of 6.46 and a *p* value of 0.002. The effect size was moderate (η^2^ = 0.05), suggesting that work context has a notable impact on perceived skills. Welch’s test confirmed this finding (*p* = 0.003).

Psychologists in palliative care support teams (M = 3.84, SD = 1.21) reported statistically significantly higher skills ratings compared to those not in such teams (M = 3.00, SD = 1.20). The *t*-test yielded a t value of 2.92 with a two-tailed *p* value of 0.004, indicating statistical significance. The effect size was moderate (Cohen’s *d* = 0.70), suggesting that working in palliative care is meaningfully associated with higher self-assessed skills.

Psychologists with prior involvement in euthanasia (M = 3.53, SD = 1.06) rated their skills statistically significantly higher than those without such experience (M = 2.28, SD = 1.07). The *t*-test yielded a t value of 8.84 and a *p* value of <0.001 for both one- and two-tailed tests, indicating a highly significant difference. The effect size was large (Cohen’s *d* = 1.18), showing that prior involvement has a strong positive association with perceived skills in dealing with euthanasia requests.

A multiple linear regression was conducted to explore the factors associated with responses to item “I have sufficient skills to handle a patient’s euthanasia request”. Independent variables comprised age categories, gender, care setting, specialized center, palliative care support team and prior involvement. The model was statistically significant and explained approximately 29.3% of the variance (R^2^ = 0.293, Adj. R^2^ = 0.275, F(6, N) = 16.26, *p* < 0.001). Significant predictors within this model included age (β = 0.16, *p* = 0.018) and prior involvement (β = −1.14, *p* < 0.001). The level of care setting showed a trend towards significance (β = 0.20, *p* = 0.055). Gender, palliative care support team, and specialized center were not significantly related to the outcome.

## 4. Discussion

This study surveyed 242 clinical psychologists in Flanders regarding attitudes, roles, and competencies related to UMS euthanasia. On average, attitude scores (M = 79.13) were significantly higher than the neutral midpoint (63), indicating strong support for patient self-determination and euthanasia as a viable option for mental suffering. These findings align with prior studies among nursing students and psychiatric nurses in Flanders ([9]; [11], [12]).

All thematic domains showed scores above the neutral midpoint, and no significant effects of age, gender, healthcare echelon, psychiatric or palliative care work, or prior euthanasia involvement on attitudes were detected. However, the absence of associations may reflect limited sample variability or measurement constraints. Broad consensus was observed, with 88% supporting self-determination, 83.9% endorsing societal acceptance of UMS euthanasia, and 90.9% viewing it as a dignified death aligning with findings by [15] ([15]) and [23] ([23]).

In practical terms, 71.9% of respondents agreed that age should influence euthanasia eligibility, consistent with Belgian law restricting euthanasia for minors to terminal conditions ([3]). A large majority (87.2%) identified themselves as crucial contributors to decision-making processes related to euthanasia requests, particularly emphasizing their role in collaboratively exploring alternatives prior to a final decision—endorsed by 96.3%. Furthermore, 88.0% felt responsible for ensuring that patients are fully informed about the nature and consequences of euthanasia, underscoring their role in patient education and support. Nearly all respondents (98.8%) advocated for a multidisciplinary approach, reflecting the complex ethical and clinical landscape of euthanasia care. Although opinions varied regarding whether psychologists are better positioned than physicians to assess euthanasia eligibility, this diversity underscores the need for clear role definitions. Psychologists were divided on attending euthanasia administration, but a majority (71.1%) considered family aftercare part of their remit. Ethical considerations were prominent, with over three-quarters supporting the right to transfer care in cases of conscientious objection. These findings support the formal inclusion of psychologists in areas such as decisional capacity assessment, collaborative decision-making, patient education, family support, and ethical consultation, consistent with recommendations highlighting psychological expertise in capacity evaluations and ethical deliberations in assisted dying contexts ([16]).

Despite overall support, it is important to situate these findings within the broader, more heterogeneous landscape of healthcare professionals’ attitudes toward (UMS) euthanasia. The literature reveals substantial variability and conflicting perspectives among nurses and physicians. Many nurses demonstrate ambivalence, shaped by ethical concerns, religious beliefs, and professional commitments to palliative care, resulting in hesitation or opposition to euthanasia participation ([7]). Physicians also exhibit mixed attitudes; a considerable proportion remain reluctant due to moral dilemmas, potential legal risks, or doubts about palliative alternatives ([1]). Among psychiatrists, recent Belgian studies indicate majority support for UMS euthanasia in adults with psychiatric conditions, accompanied by notable reservations about active involvement, reflecting concerns related to legality, ethics, and therapeutic relationships ([28]). These divergent findings highlight the ethical complexity and cultural factors shaping professional perspectives. Given the sensitive nature of UMS euthanasia, it is essential to acknowledge the moral distress and ethical tensions experienced by clinical psychologists themselves. Moral distress arises when clinicians perceive conflicts between their professional values—such as preserving life—and the obligation to participate in assisted dying, a phenomenon well documented among healthcare workers involved in euthanasia practices ([24]). Psychologists may face significant internal conflict, balancing respect for patient autonomy with their own moral and professional commitments. The present finding—that over three-quarters of participants support the right to transfer care due to conscientious objection—reflects an important ethical safeguard protecting clinicians’ moral integrity and mental health. Structured ethical supervision, peer support, and clear institutional policies are vital to mitigating moral distress and sustaining psychological well-being among practitioners engaged in these challenging cases.

An important tension emerged between the predominantly positive attitudes clinical psychologists expressed toward UMS euthanasia and their reported lack of sufficient knowledge and practical skills to manage such cases effectively. While the mean attitude score indicated strong support for patient self-determination and integration of psychologists within UMS euthanasia care, nearly half of participants (47.1%) reported insufficient knowledge to handle euthanasia requests, and 37.6% indicated lacking necessary practical skills. This gap poses a potential risk to the quality and consistency of care provided to patients requesting UMS euthanasia, underscoring an urgent need for targeted education and training. Furthermore, an overwhelming majority (94.2%) felt that UMS euthanasia was inadequately covered during their formal education or training. Simulation-based learning and specialized curricular modules could help bridge this competence gap, ensuring that positive attitudes translate into clinical readiness to manage the complex ethical, legal, and psychological dimensions of UMS euthanasia. This finding aligns with prior research showing that support for euthanasia does not necessarily equate with professional preparedness, emphasizing the critical importance of capacity building within the clinical psychology workforce ([29]). Simulation-based education offers valuable experiential learning opportunities, enabling future psychologists to engage with challenging scenarios in a safe, structured environment and to develop confidence and competence under supervision ([13]; [20]).

Knowledge and skills related to euthanasia were significantly higher among psychologists working in secondary and tertiary healthcare settings, those involved in palliative care teams, and individuals with prior experience handling euthanasia requests. This pattern suggests that greater exposure to complex clinical cases, multidisciplinary collaboration, and direct involvement in end-of-life decision-making substantially enhance professional competence in this domain. Psychologists in these contexts are more likely to face challenging ethical scenarios, participate in specialized training, and develop practical expertise through ongoing interactions with patients and families experiencing psychologically unbearable suffering. In contrast, age had no measurable impact on self-reported knowledge and skills, indicating that euthanasia expertise depends less on career longevity or personal maturity and more on the specific clinical environment, relevant professional development opportunities, and experiential learning.

These findings have several policy implications. First, there is a need to formally define the roles and responsibilities of clinical psychologists in the UMS euthanasia process through standardized national guidelines. Currently, Belgium lacks clear scientific or legal definitions of their position, despite evidence showing that psychologists play a vital supportive role for patients, families, and multidisciplinary teams by addressing fears, needs, and expectations ([18]; [30]). Establishing formal guidelines would specify when and how psychologists are involved and clarify their tasks, thereby improving integration into UMS euthanasia procedures and fostering consistent, coordinated care across settings. Such guidance should particularly address psychologists’ functions in euthanasia within palliative care contexts, where neither legislation ([10]) nor professional frameworks ([22]) currently define their role. These guidelines would especially benefit psychologists with limited experience in euthanasia by equipping them with clear protocols and practical tools to enhance their contribution to patient and family support. Notably, age was positively associated with responses to the items “I have sufficient skills to handle a patient’s euthanasia request” and “I have sufficient knowledge to handle a patient’s euthanasia request,” indicating that older professionals tend to express more favorable or distinct views on these constructs. Second, there is a clear need to strengthen education and training for clinical psychologists on UMS euthanasia. The reported gaps in knowledge and skills, combined with the near-universal perception of insufficient coverage in current curricula, highlight an opportunity for reform. Greater curricular emphasis on euthanasia, supported by simulation-based training, could improve clinical preparedness, build professional confidence, and enhance patient care. These measures align with similar improvements advocated in nursing education, such as simulation training ([13]). Our recommendations for clearer guidelines and enhanced simulation-based training for clinical psychologists on UMS euthanasia align with international efforts in countries like the Netherlands and Canada. In the Netherlands, where UMS euthanasia has been legal since 2002, physicians and nurse practitioners undergo rigorous training aligned with strict legal criteria and due care procedures, supported by institutions such as the Expertisecentrum Euthanasie, which provides ongoing counseling, training, and expertise sharing for clinicians handling complex cases. In Canada, clinician education for (UMS) euthanasia, focuses on understanding legal criteria and safeguards, enhancing skills in capacity assessments specific to mental illness, navigating ethical challenges balancing autonomy and protection, and developing communication abilities for sensitive discussions. This training is delivered through accredited national curricula, such as the Canadian MAID Curriculum developed by the Canadian Association of MAID Assessors and Providers, comprising multiple modules offered via workshops and online sessions to licensed physicians and nurse practitioners nationwide. While Belgium, Canada and the Netherlands have developed nationally accredited, comprehensive training programs for physicians and nurse practitioners involved in (UMS) euthanasia, there are currently no specialized training programs explicitly designed for clinical psychologists.

In practical terms, implementing these recommendations will require coordinated action among policymakers, professional associations, educational institutions, and healthcare providers. Universities and postgraduate programs should integrate structured UMS euthanasia modules into clinical psychology curricula, including mandatory simulation training on complex case scenarios. Healthcare institutions need to implement clear internal protocols defining psychologists’ roles within multidisciplinary euthanasia procedures to ensure uniformity across settings. Professional associations, such as the Flemish Association of Clinical Psychologists (VVKP), should collaborate with policymakers to develop and disseminate standardized practice guidelines.

This study has several limitations that should be considered when interpreting the findings. First, the cross-sectional design provides only a snapshot of attitudes and competencies at a single point in time, limiting insight into how these may evolve. Second, the exclusive use of self-report questionnaires may introduce response biases, particularly social desirability bias, which could inflate positive attitudes or perceived competencies. Additionally, discrepancies may exist between participants’ self-reported attitudes and their actual clinical practices, as intentions do not always translate into behavior. Third, the sampling strategy relied heavily on professional associations within Flanders, potentially introducing selection bias. Members of these associations may be more professionally engaged or possess greater interest or expertise in euthanasia, which could limit the representativeness of our sample and constrain the generalizability of findings to the wider population of clinical psychologists in the region. Our sample of 242 actively practicing clinical psychologists represents only a small portion of the estimated total registered population. Although this constitutes a significant proportion, the response rate could not be precisely calculated due to reliance on organizational sharing and voluntary participation. Fourth, important factors such as personal values, cultural or religious beliefs, and previous experiences with euthanasia were not measured but might have influenced responses. Previous research has demonstrated that religiosity and cultural background strongly shape healthcare professionals’ attitudes toward euthanasia ([14]). Religious beliefs often provide fundamental ethical frameworks influencing acceptance or opposition to euthanasia, with more religious individuals tending to hold more negative attitudes. These cultural and religious dimensions are critical for fully understanding professional perspectives and should be incorporated in future studies. Fifth, although the survey instrument was adapted from measures previously validated in nursing student populations—showing good psychometric properties—it was not subjected to a separate pilot study or additional reliability testing within this professional group. Future validation efforts should focus on establishing this instrument’s reliability and validity specifically among psychologists. Finally, future research employing longitudinal, mixed-method designs and larger, more diverse samples would deepen understanding of the complex and sensitive issues surrounding psychologists’ roles in UMS euthanasia.

## 5. Conclusions

This study provides novel empirical insights into the attitudes, roles, and competencies of clinical psychologists in Flanders regarding UMS euthanasia. The findings reveal broad professional endorsement of patient self-determination and strong support for integrating clinical psychologists within multidisciplinary euthanasia care teams. Nevertheless, significant gaps in knowledge, practical skills, and educational preparation were identified. Addressing these deficiencies through clearer role delineations, the development of standardized national guidelines, and enhanced training programs—including simulation-based learning—could enhance both professional competency and the quality of care offered to patients and their families. Although findings are based on a heterogeneous sample of psychologists from diverse care settings, their generalizability is limited by the study’s design and sampling framework. Therefore, further research using larger, more representative samples and qualitative methods is warranted to comprehensively clarify clinical psychologists’ roles in UMS euthanasia and to inform policy and practice at national and international levels.

## Figures and Tables

**Figure 1 ejihpe-15-00228-f001:**
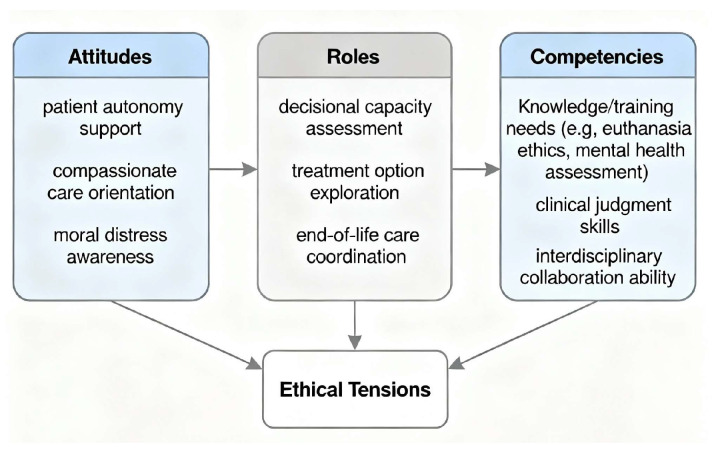
Conceptual Model of Attitudes, Roles, and Competencies of Clinical Psychologists Regarding UMS Euthanasia.

**Table 1 ejihpe-15-00228-t001:** Attitudes of clinical psychologists regarding UMS euthanasia (*n* = 242).

		Strongly Disagree	Disagree	Neither Agree nor Disagree	Agree	Strongly Agree
	Ethical Considerations	*n* (%)	*n* (%)	*n* (%)	*n* (%)	*n* (%)
1a	A person with a mental illness has the right to decide to die.	4 (1.7%)	4 (1.7%)	21 (8.7%)	153 (63.2%)	60 (24.8%)
1b	Inducing death for merciful reason is wrong.	29 (12.0%)	69 (28.5%)	72 (29.8%)	43 (17.8%)	29 (12.0%)
1c	UMS euthanasia should be accepted in today’s society.	5 (2.1%)	11 (4.5%)	23 (9.5%)	134 (55.4%)	69 (28.5%)
1d	There are never cases when UMS-euthanasia is appropriate.	125 (51.7%)	100 (41.3%)	9 (3.7%)	4 (1.7%)	4 (1.7%)
1e	UMS euthanasia is helpful at the right time and place.	6 (2.5%)	7 (2.9%)	18 (7.4%)	124 (51.2%)	87 (36.0%)
1f	UMS euthanasia is a human act.	4 (1.7%)	6 (2.5%)	40 (16.5%)	126 (52.1%)	66 (27.3%)
1g	UMS euthanasia should be against the law.	145 (59.9%)	78 (32.2%)	8 (3.3%)	8 (3.3%)	3 (1.2%)
1h	UMS euthanasia should be applied when a mentally ill person is out of treatment.	28 (11.6%)	54 (22.3%)	67 (27.7%)	65 (26.9%)	28 (11.6%)
1i	The taking of human life is wrong no matter what the circumstances.	138 (57.0%)	86 (35.5%)	13 (5.4%)	4 (1.7%)	1 (0.4%)
1j	UMS euthanasia is acceptable in cases when all hope of recovery is gone.	4 (1.7%)	22 (9.7%)	40 (16.5%)	115 (47.5%)	61 (25.2%)
1k	UMS euthanasia gives a person a chance to die with dignity.	3 (1.2%)	6 (2.5%)	13 (5.4%)	103 (42.6%)	117 (48.3%)
	Treasuring life					
2a	The age of the patient should play a role in euthanasia, for example in distinguishing between minor and adult patients.	6 (2.5%)	29 (12.0%)	33 (13.6%)	119 (49.2%)	55 (22.7%)
2b	If a mentally incurable person is increasingly concerned about the burden that his or her deterioration of health has placed on his or her family, I will support his or her request for euthanasia.	34 (14.0%)	80 (33.1%)	97 (40.1%)	23 (9.5%)	8 (3.3%)
2c	UMS euthanasia will lead to abuses.	39 (16.1%)	102 (42.1%)	76 (31.4%)	21 (8.7%)	4 (1.7%)
2d	I have faith in the Belgian medical system to implement UMS euthanasia properly.	14 (5.8%)	33 (13.6%)	56 (23.1%)	111 (45.9%)	28 (11.6%)
	Practical considerations					
3a	There are very few cases when UMS euthanasia is acceptable.	37 (15.3%)	83 (34.3%)	58 (24.0%)	51 (21.1%)	13 (5.4%)
3b	UMS euthanasia should be practiced only to eliminate physical pain and not mental suffering.	131 (54.1%)	95 (39.3%)	7 (2.9%)	4 (1.7%)	5 (2.1%)
3c	One’s job is to sustain and preserve life, not to end it.	60 (24.8%)	77 (31.8%)	52 (21.5%)	46 (19.0%)	7 (2.9%)
3d	One of the key professional ethics of physicians is to prolong lives, not to end lives.	60 (24.8%)	89 (36.8%)	68 (28.1%)	22 (9.1%)	3 (1.2%)
	Naturalistic beliefs					
4a	A person should not be kept alive by machines.	10 (4.1%)	17 (7.0%)	95 (39.3%)	88 (36.4%)	32 (13.2%)
4b	Natural death is a cure for suffering.	30 (12.4%)	51 (21.1%)	112 (46.3%)	46 (19.0%)	3 (1.2%)

**Table 3 ejihpe-15-00228-t003:** Role of Clinical Psychologists Regarding UMS Euthanasia (*n* = 242).

		Strongly Disagree	Disagree	Neither Agree nor Disagree	Agree	Strongly Agree
		*n* (%)	*n* (%)	*n* (%)	*n* (%)	*n* (%)
1	Psychologists have an important role in the decision-making process regarding euthanasia for patients experiencing unbearable mental suffering.	2 (0.8%)	12 (5.0%)	17 (7.0%)	129 (53.3%)	82 (33.9%)
2	Psychologists should support the patient in exploring alternatives to euthanasia before a final decision is made.	1 (0.4%)	2 (0.8%)	6 (2.5%)	81 (33.5%)	152 (62.8%)
3	Psychologists should ensure that the patient is sufficiently informed about the nature and consequences of euthanasia.	2 (0.8%)	10 (4.1%)	17 (7.0%)	79 (32.6%)	134 (55.4%)
4	A euthanasia request should be addressed using a multidisciplinary approach.	1 (0.4%)	-	2 (0.8%)	23 (9.5%)	216 (89.3%)
5	It is the psychologist’s role to accompany the patient during the actual performance of euthanasia.	25 (10.3%)	42 (17.4%)	98 (40.5%)	56 (23.1%)	21 (8.7/%)
6	It is the psychologist’s role to provide support to the patient’s family and close relatives after the patient’s death.	4 (1.7%)	18 (7.4%)	48 (19.8%)	121 (50.0%)	51 (21.1%)
7	A psychologist who has ethical objections to euthanasia should be able to transfer the patient’s care to another colleague.	6 (2.5%)	15 (6.2%)	36 (14.9%)	89 (36.8%)	96 (39.7%)
8	A psychologist, due to their deeper knowledge of the patient, is better placed than a physician to assess a euthanasia request.	13 (5.4%)	73 (30.2%)	91 (37.6%)	49 (20.2%)	16 (6.6%)
9	Psychologists should be members of the Federal Control and Evaluation Commission for Euthanasia (FCEE).	3 (1.2%)	5 (2.1%)	41 (16.9%)	110 (45.5%)	83 (34.3%)

**Table 4 ejihpe-15-00228-t004:** Competencies of Clinical Psychologists Regarding UMS Euthanasia (*n* = 242).

		Strongly Disagree	Disagree	Neither Agree nor Disagree	Agree	Strongly Agree
		*n* (%)	*n* (%)	*n* (%)	*n* (%)	*n* (%)
1	I have sufficient knowledge to handle a patient’s euthanasia request.	35 (14.5%)	79 (32.6%)	45 (18.6%)	61 (25.2%)	22 (9.1%)
2	I have sufficient skills to handle a patient’s euthanasia request.	28 (11.6%)	63 (26.0%)	42 (17.4%)	83 (34.3%)	26 (10.7%)
3	UMS euthanasia was sufficiently addressed during my education.	160 (66.1%)	68 (28.1%)	4 (1.7%)	4 (1.7%)	6 (2.5%)
4	More attention should be devoted to the topic of “euthanasia” in clinical psychology education.	3 (1.2%)	5 (2.1%)	19 (7.9%)	103 (42.6%)	112 (46.3%)
5	More attention should be devoted to simulation-based lessons regarding UMS euthanasia in psychology education.	3 (1.2%)	7 (2.9%)	62 (25.6%)	84 (34.7%)	86 (35.5%)

**Table 2 ejihpe-15-00228-t002:** Attitudes of clinical psychologists regarding UMS euthanasia by theme (*n* = 242).

	N	M	SD	Min	Max
Ethical Considerations	242	43.96	6.63	11.00	55.00
Treasuring life	242	15.06	3.13	4.00	20.00
Practical considerations	242	13.39	2.19	7.00	19.00
Naturalistic beliefs	242	6.72	1.56	2.00	10.00
TOTAL	242	79.13	10.54	29.00	103.00

## Data Availability

The original data presented in the study are openly available in Mendeley Data at https://doi.org/10.17632/23jm4wbzxv.2 (accessed on 26 August 2025).

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
