# Peer review of "Attitudes, Roles, and Competencies of Clinical Psychologists Regarding Euthanasia Due to Unbearable Mental Suffering"

_ejihpe, 2025, doi:10.3390/ejihpe15110228_

Round 1
Reviewer 1 Report
Comments and Suggestions for Authors
This article addresses a highly sensitive and socially debated topic: the attitudes, roles, and competencies of clinical psychologists in relation to euthanasia due to unbearable mental suffering in Belgium. The study is timely and relevant, as it contributes to ongoing discussions about the ethical, professional, and legal frameworks surrounding assisted dying, especially in cases of psychiatric suffering. The paper provides valuable empirical data and offers insights that could inform both practice and policy. At the same time, certain academic aspects of the manuscript could be strengthened to improve clarity.
General comment: The manuscript does not clearly explain how "unbearable mental suffering" is operationalized or assessed, which represents a significant methodological limitation compared to the more objective criteria used in somatic end-of-life care.
Introduction:
Add A sharper articulation of the knowledge gap in the existing literature (why exactly are psychologists’ attitudes under-researched compared to, for example, physicians or nurses?).
A clearer definition of “unbearable mental suffering” and how it has been operationalized in previous studies is missing.
A more explicit link between the international debate and the Belgian context- how generalizable are the findings expected to be?
The paper lacks clearly stated hypotheses.
Methods:
Add more detail on questionnaire validation: was the adapted UMS-EAS-NL scale piloted or tested for psychometric reliability beyond referencing earlier studies?
How representative is the sample compared to the total population of psychologists in Flanders? What was the response rate?
Line 122: Add subheading: "Data Analyses"
Discussion:
Add a deeper engagement with conflicting literature (e.g., studies showing more critical or hesitant attitudes among other professional groups).
Consideration of cultural or religious factors that might shape attitudes but were not measured in this study is missing. See: https://doi.org/10.3390/ijerph18126396
Conclusions:
Practical recommendations could be more specific (e.g., curriculum design, interdisciplinary training programs).
Future studies could compare professionals across different disciplines, countries, and religious backgrounds, especially between contexts where euthanasia is legally permitted and those where it is strictly prohibited, in order to better capture the depth and diversity of perspectives in this contentious field.
Author Response
Comment 1: The manuscript does not clearly explain how "unbearable mental suffering" is operationalized or assessed, which represents a significant methodological limitation compared to the more objective criteria used in somatic end-of-life care.
Response: Thank you for highlighting the importance of clarifying how "unbearable mental suffering" is operationalized in our study. We have now included a detailed explanation, specifying that UMS is understood in accordance with the Belgian Euthanasia Act (2002), in the Methods.
Changes: In this study, UMS is conceptualized in accordance with the criteria established by the Belgian Euthanasia Act (2002). UMS refers to a subjective experience of severe psychological distress that the patient perceives as unbearable and is considered incurable and untreatable following comprehensive multidisciplinary evaluation. In contrast to somatic suffering—which can often be substantiated by objective medical findings—the assessment of UMS relies primarily on clinical judgment, patient self-report, and the integration of psychiatric diagnoses and existential dimensions. Within the structured Belgian legal framework, specialized clinicians are required to conduct detailed evaluations of the patient’s mental state, identify underlying conditions, and determine whether the experienced suffering meets the legal and clinical thresholds for UMS euthanasia.
Comment 2: Add A sharper articulation of the knowledge gap in the existing literature (why exactly are psychologists’ attitudes under-researched compared to, for example, physicians or nurses?).
Response: Thank you for this important observation. We have revised the introduction to more clearly articulate the existing knowledge gap concerning clinical psychologists’ attitudes toward UMS euthanasia.
Changes: Although substantial research has explored physicians’ and nurses’ attitudes toward euthanasia, traditionally examined due to their direct involvement in physical care, empirical insight into clinical psychologists’ perspectives remains limited (De Hert et al., 2015; Verhofstadt et al., 2020). Yet, clinical psychologists hold a distinctive position in this context: they assess decisional capacity, evaluate UMS, and contribute to multidisciplinary end-of-life deliberations. Their psychological expertise is crucial to ensuring ethically sound decision-making, but their roles and attitudes have received little systematic investigation. Understanding their views and professional competencies is essential for developing integrative care models that balance ethical sensitivity with psychosocial expertise.
Comment 3: A clearer definition of “unbearable mental suffering” and how it has been operationalized in previous studies is missing.
Response: Thank you for highlighting the need for a clearer definition and operationalization of “unbearable mental suffering” (UMS). We have now incorporated an explicit explanation in the manuscript that defines UMS in accordance with the Belgian Euthanasia Act (2002) and related empirical literature.
Changes: Within Belgium, euthanasia may also be granted in cases of unbearable mental suffering (UMS) arising from non-terminal or psychiatric disorders considered both untreatable and hopeless (Gallego et al., 2022). UMS encompasses a wide spectrum of psychological distress, from severe psychiatric illness to existential despair without an identifiable medical diagnosis. It is a deeply subjective and complex phenomenon, relying primarily on patients’ self-reports and clinical interpretation rather than objective biomarkers. Its evaluation requires careful multidisciplinary deliberation involving both medical and mental health experts. Therefore, under the Belgian Euthanasia Act (2002), specific legal criteria must be satisfied: the patient must be an adult and competent, experiencing a medically hopeless situation resulting in unbearable physical and/or mental suffering caused by an incurable condition or accident, and the euthanasia request must be voluntary, well-considered, and repeated. The attending physician is required to fully inform the patient about their health status, potential therapies, prognosis, and palliative options, and must consult a second, independent physician. In non-terminal or psychiatric cases, a third, specialized physician must be consulted, and a one-month reflection period is mandatory (Belgian Ministry of Justice, 2002).
Comment 4: A more explicit link between the international debate and the Belgian context- how generalizable are the findings expected to be?
Response: Thank you for this insightful suggestion. We have revised the introduction to explicitly situate our study within the Belgian euthanasia framework, which is one of the most established and comprehensive regulatory contexts for UMS euthanasia. We acknowledge that while the mature Belgian legal and healthcare system provides a unique backdrop for our findings, the generalizability to other countries is limited by differences in legislation, cultural attitudes, and healthcare structures.
Changes: While the euthanasia debate is global, Belgium provides a particularly instructive legal and healthcare context, featuring a mature regulatory framework and established multidisciplinary procedures. Conducting this study in Belgium therefore offers a valuable opportunity to explore psychologists’ professional views within one of the world’s most developed systems of (UMS) euthanasia governance. Findings are expected to enrich international understanding despite contextual differences across countries. Guided by these considerations, the present study addresses the following question: What are the attitudes, roles, and competencies of clinical psychologists regarding UMS euthanasia?
Comment 5: The paper lacks clearly stated hypotheses.
Response: Thank you for your valuable comment. In response, we have included a clear set of hypotheses in the introduction section to enhance the conceptual clarity and focus of the study.
Changes: Based on prior research and identified knowledge gaps, the study tests the following hypotheses:
- Clinical psychologists in Flanders will generally exhibit positive attitudes toward UMS euthanasia, reflecting support for patient self-determination and professional involvement in euthanasia care.
- Attitudes, perceived roles, and self-reported competencies will vary according to demographic (e.g., age, gender) and professional factors (e.g., healthcare echelon, prior experience with euthanasia requests).
- Despite positive attitudes, psychologists will report notable gaps in knowledge and skills, highlighting the need for enhanced education and training initiatives.
Comment 6: Add more detail on questionnaire validation: was the adapted UMS-EAS-NL scale piloted or tested for psychometric reliability beyond referencing earlier studies?
Response: Thank you for your comment regarding the validation of the adapted UMS-EAS-NL questionnaire. Our study builds upon the previous thorough validation carried out within nursing student populations, where the scale demonstrated good psychometric properties, including strong internal consistency and construct validity. However, we did not conduct a separate pilot study or perform additional reliability testing specifically within the clinical psychologist sample. We acknowledge this as a limitation and suggest that future research might further validate the instrument in this specific professional group.
Changes: Fifth, although the survey instrument was adapted from measures previously validated in nursing student populations—showing good psychometric properties—it was not subjected to a separate pilot study or additional reliability testing within this professional group. Future validation efforts should focus on establishing this instrument’s reliability and validity specifically among psychologists.
Comment 7: How representative is the sample compared to the total population of psychologists in Flanders? What was the response rate?
Response: The survey was disseminated through professional organizations, reaching over 2,500 members, which constitutes a significant proportion of the regional population. While we do not have exact response rates due to the nature of broad dissemination via email invitations and organizational sharing, the recruitment strategy aimed at maximizing participation within the target population. Nonetheless, the representativeness of our sample relative to the total population of clinical psychologists in Flanders should be interpreted with caution, and future research could benefit from more precise response rate calculations or sampling strategies to better quantify this aspect.
Changes: This study has several limitations that should be considered when interpreting the findings. First, the cross-sectional design provides only a snapshot of attitudes and competencies at a single point in time, limiting insight into how these may evolve. Second, the exclusive use of self-report questionnaires may introduce response biases, particularly social desirability bias, which could inflate positive attitudes or perceived competencies. Additionally, discrepancies may exist between participants’ self-reported attitudes and their actual clinical practices, as intentions do not always translate into behavior. Third, the sampling strategy relied heavily on professional associations within Flanders, potentially introducing selection bias. Members of these associations may be more professionally engaged or possess greater interest or expertise in euthanasia, which could limit the representativeness of our sample and constrain the generalizability of findings to the wider population of clinical psychologists in the region. Our sample of 242 actively practicing clinical psychologists represents only a small portion of the estimated total registered population. Although this constitutes a significant proportion, the response rate could not be precisely calculated due to reliance on organizational sharing and voluntary participation.
Comment 8: Line 122: Add subheading: "Data Analyses"
Response: Thank you for this helpful suggestion. We have added the subheading "Data Analyses" to improve the clarity and organization of the Methods section.
Comment 9: Add a deeper engagement with conflicting literature (e.g., studies showing more critical or hesitant attitudes among other professional groups).
Response: Thank you for this valuable suggestion. We have expanded the discussion to more thoroughly engage with conflicting findings in the literature regarding attitudes toward UMS euthanasia among various healthcare professionals.
Changes: All thematic domains showed scores above the neutral midpoint, and no significant effects of age, gender, healthcare echelon, psychiatric or palliative care work, or prior euthanasia involvement on attitudes were detected. However, the absence of associations may reflect limited sample variability or measurement constraints. Broad consensus was observed, with 88% supporting self-determination, 83.9% endorsing societal acceptance of UMS euthanasia, and 90.9% viewing it as a dignified death aligning with findings by Fontalis et al. (2018) and Picón-Jaimes et al. (2022).
Despite overall support, it is important to situate these findings within the broader, more heterogeneous landscape of healthcare professionals’ attitudes toward (UMS) euthanasia. The literature reveals substantial variability and conflicting perspectives among nurses and physicians. Many nurses demonstrate ambivalence, shaped by ethical concerns, religious beliefs, and professional commitments to palliative care, resulting in hesitation or opposition to euthanasia participation (Cayetano-Penman et al., 2021). Physicians also exhibit mixed attitudes; a considerable proportion remain reluctant due to moral dilemmas, potential legal risks, or doubts about palliative alternatives (Archer et al., 2025). Among psychiatrists, recent Belgian studies indicate majority support for UMS euthanasia in adults with psychiatric conditions, accompanied by notable reservations about active involvement, reflecting concerns related to legality, ethics, and therapeutic relationships (Verhofstadt et al., 2020). These divergent findings highlight the ethical complexity and cultural factors shaping professional perspectives. Given the sensitive nature of UMS euthanasia, it is essential to acknowledge the moral distress and ethical tensions experienced by clinical psychologists themselves. Moral distress arises when clinicians perceive conflicts between their professional values—such as preserving life—and the obligation to participate in assisted dying, a phenomenon well documented among healthcare workers involved in euthanasia practices (Pinto et al., 2025). Psychologists may face significant internal conflict, balancing respect for patient autonomy with their own moral and professional commitments. The present finding—that over three-quarters of participants support the right to transfer care due to conscientious objection—reflects an important ethical safeguard protecting clinicians’ moral integrity and mental health. Structured ethical supervision, peer support, and clear institutional policies are vital to mitigating moral distress and sustaining psychological well-being among practitioners engaged in these challenging cases.
Comment 10: Consideration of cultural or religious factors that might shape attitudes but were not measured in this study is missing. See: https://doi.org/10.3390/ijerph18126396
Response: We thank the reviewer for highlighting this important aspect. We have now acknowledged this limitation explicitly in the limitations section of the manuscript.
Changes: Fourth, important factors such as personal values, cultural or religious beliefs, and previous experiences with euthanasia were not measured but might have influenced responses. Previous research has demonstrated that religiosity and cultural background strongly shape healthcare professionals’ attitudes toward euthanasia (Dopelt et al., 2021). Religious beliefs often provide fundamental ethical frameworks influencing acceptance or opposition to euthanasia, with more religious individuals tending to hold more negative attitudes. These cultural and religious dimensions are critical for fully understanding professional perspectives and should be incorporated in future studies.

Reviewer 2 Report
Comments and Suggestions for Authors
This is a timely and relevant article addressing the attitudes, roles, and competencies of clinical psychologists in Belgium regarding euthanasia requests due to unbearable mental suffering (UMS). The subject is highly significant given the ongoing ethical, clinical, and societal debates around medically assisted dying, particularly in psychiatric contexts. The authors provide empirical data from a sample of 242 clinical psychologists in Flanders, applying a cross-sectional survey methodology.
The manuscript is well written, clearly structured, and includes comprehensive analyses. The findings, particularly the high general support among psychologists for UMS euthanasia, the perceived importance of their role, and the strong consensus about insufficient training, are valuable contributions to the literature.
However, while the study is solid, several aspects require further clarification and strengthening before publication.
Major Comments
A.Theoretical and Conceptual Framing
- The introduction situates the study in the legal and policy context but could benefit from a deeper engagement with theoretical frameworks regarding professional ethics, autonomy, and the psychological dimensions of suffering. For example, how do debates on autonomy vs. protection of vulnerable individuals frame the role of psychologists?
- While the literature review is broad, it is somewhat descriptive. A more critical positioning of existing research (e.g., highlighting gaps specifically concerning psychologists, rather than mainly nurses and physicians) would sharpen the contribution.
B. Methodology
- Sampling and Representativeness: The recruitment strategy relies heavily on professional associations, which may bias toward more engaged or specialized psychologists. The authors should discuss possible selection bias and how it may affect generalizability.
- Instrument Validation: The questionnaire (UMS-EAS-NL) is adapted from studies with nursing students. More detail is needed on how items were modified for psychologists and whether any pilot testing was conducted to ensure construct validity in this new professional group. A Confirmatory Factor Analysis could be useful to assess if the instrument’s structure is confirmed by the study’s participants. Reliability measures would also be welcome.
- Cross-sectional Limits: The design only captures self-reported attitudes at one point in time. The authors should acknowledge limitations regarding causality, social desirability bias, and potential discrepancies between reported attitudes and actual practice.
C. Results and Interpretation
- The statistical analyses are thorough, but the interpretation at times overstates the implications. For example, the non-significant role of demographics in shaping attitudes should be contextualized as possibly due to sample composition or measurement limitations.
- The discussion would benefit from contrasting the strong positive attitudes with the reported lack of competencies. This tension could be highlighted more explicitly: psychologists largely support UMS euthanasia but do not feel adequately prepared, creating a potential gap in care.
D. Policy and Training Implications
- The recommendations for clearer guidelines and simulation-based training are valuable but would be stronger if linked to international examples. For instance, how have training programs in the Netherlands or Canada addressed similar issues?
- The call for standardized roles is important but currently presented in broad terms. The authors might suggest specific domains where psychologists could be formally included (e.g., assessment of capacity, family aftercare, supervision of ethical reflection).
E. Ethical Considerations
- Given the sensitivity of the topic, a more explicit reflection on ethical tensions for psychologists themselves (e.g., moral distress, conscientious objection) would enrich the discussion. This is only briefly touched upon in the results.
Minor Comments
A. Clarity and Style
- The manuscript is generally well written, but some sections could be more concise, particularly the presentation of results where extensive numerical details might be better summarized in text while leaving the full data in tables.
- Ensure consistent terminology: sometimes “UMS euthanasia” is used, other times simply “euthanasia.” Clarify at the outset and use consistently.
B. Tables and Figures
- A conceptual diagram summarizing the relationship between attitudes, roles, and competencies could help readers grasp the study framework.
C. References
- The reference list is up to date and relevant, but a few important philosophical and ethical sources (e.g., Beauchamp & Childress on biomedical ethics, discussions of autonomy in psychiatry, etc.) could strengthen the theoretical grounding.
In summary, the article makes an important and original contribution to understanding the perspectives of clinical psychologists on UMS euthanasia. However, revisions are required to (1) strengthen theoretical framing, (2) clarify methodological rigor and limitations, (3) contextualize results within broader ethical debates, and (4) refine policy/training implications with more specificity.
Author Response
Comment 1: The introduction situates the study in the legal and policy context but could benefit from a deeper engagement with theoretical frameworks regarding professional ethics, autonomy, and the psychological dimensions of suffering. For example, how do debates on autonomy vs. protection of vulnerable individuals frame the role of psychologists?
Response: Thank you for your constructive suggestion regarding the theoretical framing in our introduction. We have now expanded the introduction to include a discussion on the ethical landscape surrounding euthanasia for unbearable mental suffering.
Changes: The ethical challenges surrounding UMS euthanasia highlight a central tension in professional psychology: reconciling respect for patient autonomy with the duty to protect vulnerable individuals. Psychologists operate at this intersection, committed to supporting self-determination and alleviating mental suffering while guarding against decisions influenced by impaired judgment or fluctuating mental states. Bioethical and psychological frameworks emphasize the importance of balancing autonomy and protection, recognizing that the subjective nature of suffering complicates assessments of capacity and consent (Beauchamp & Childress, 2019). Situating the present study within these frameworks underscores the need to clarify clinical psychologists’ roles, ethical orientations, and required competencies regarding UMS euthanasia.
Comment 2: While the literature review is broad, it is somewhat descriptive. A more critical positioning of existing research (e.g., highlighting gaps specifically concerning psychologists, rather than mainly nurses and physicians) would sharpen the contribution.
Response: Thank you for this valuable suggestion. We expanded the introduction by critically positioning the existing literature, emphasizing that while research has extensively examined physicians’ and nurses’ attitudes and roles in euthanasia, the perspectives of clinical psychologists remain markedly underexplored.
Changes:
Although substantial research has explored physicians’ and nurses’ attitudes toward euthanasia, traditionally examined due to their direct involvement in physical care, empirical insight into clinical psychologists’ perspectives remains limited (De Hert et al., 2015; Verhofstadt et al., 2020). Yet, clinical psychologists hold a distinctive position in this context: they assess decisional capacity, evaluate UMS, and contribute to multidisciplinary end-of-life deliberations. Their psychological expertise is crucial to ensuring ethically sound decision-making, but their roles and attitudes have received little systematic investigation. Understanding their views and professional competencies is essential for developing integrative care models that balance ethical sensitivity with psychosocial expertise.
Comment 3: Sampling and Representativeness: The recruitment strategy relies heavily on professional associations, which may bias toward more engaged or specialized psychologists. The authors should discuss possible selection bias and how it may affect generalizability.
Response: Thank you for this insightful observation. We acknowledge that recruiting participants primarily through professional associations may introduce selection bias, as members of these organizations might be more professionally engaged, have greater interest or expertise in euthanasia, or differ in other relevant characteristics from non-members. This may limit the representativeness of our sample and constrain the generalizability of our findings to the broader population of clinical psychologists in Flanders. We have addressed this potential limitation explicitly in the revised manuscript’s limitations section.
Changes: Third, the sampling strategy relied heavily on professional associations within Flanders, potentially introducing selection bias. Members of these associations may be more professionally engaged or possess greater interest or expertise in euthanasia, which could limit the representativeness of our sample and constrain the generalizability of findings to the wider population of clinical psychologists in the region. Our sample of 242 actively practicing clinical psychologists represents only a small portion of the estimated total registered population. Although this constitutes a significant proportion, the response rate could not be precisely calculated due to reliance on organizational sharing and voluntary participation.
Comment 4: Instrument Validation: The questionnaire (UMS-EAS-NL) is adapted from studies with nursing students. More detail is needed on how items were modified for psychologists and whether any pilot testing was conducted to ensure construct validity in this new professional group. A Confirmatory Factor Analysis could be useful to assess if the instrument’s structure is confirmed by the study’s participants. Reliability measures would also be welcome.
Response: Thank you for your comment regarding the validation of the adapted UMS-EAS-NL questionnaire. Our study builds upon the previous thorough validation carried out within nursing student populations, where the scale demonstrated good psychometric properties, including strong internal consistency and construct validity. However, we did not conduct a separate pilot study or perform additional reliability testing specifically within the clinical psychologist sample. We acknowledge this as a limitation and suggest that future research might further validate the instrument in this specific professional group.
Changes: Fifth, although the survey instrument was adapted from measures previously validated in nursing student populations—showing good psychometric properties—it was not subjected to a separate pilot study or additional reliability testing within this professional group. Future validation efforts should focus on establishing this instrument’s reliability and validity specifically among psychologists.
Comment 5: Cross-sectional Limits: The design only captures self-reported attitudes at one point in time. The authors should acknowledge limitations regarding causality, social desirability bias, and potential discrepancies between reported attitudes and actual practice.
Response: Thank you for highlighting the inherent limitations of the cross-sectional design. We have acknowledged these in the revised manuscript.
Changes: Second, the exclusive use of self-report questionnaires may introduce response biases, particularly social desirability bias, which could inflate positive attitudes or perceived competencies. Additionally, discrepancies may exist between participants’ self-reported attitudes and their actual clinical practices, as intentions do not always translate into behavior.
Comment 6: The statistical analyses are thorough, but the interpretation at times overstates the implications. For example, the non-significant role of demographics in shaping attitudes should be contextualized as possibly due to sample composition or measurement limitations.
Response: Thank you for this important comment. We acknowledge that non-significant findings regarding demographic predictors of attitudes should be interpreted with caution. We have revised the discussion to clarify that the absence of significant associations may reflect features of our sample, such as limited variability or sample size, rather than definitive evidence of no effect.
Changes: All thematic domains showed scores above the neutral midpoint, and no significant effects of age, gender, healthcare echelon, psychiatric or palliative care work, or prior euthanasia involvement on attitudes were detected. However, the absence of associations may reflect limited sample variability or measurement constraints. Broad consensus was observed, with 88% supporting self-determination, 83.9% endorsing societal acceptance of UMS euthanasia, and 90.9% viewing it as a dignified death aligning with findings by Fontalis et al. (2018) and Picón-Jaimes et al. (2022).
Comment 7: The discussion would benefit from contrasting the strong positive attitudes with the reported lack of competencies. This tension could be highlighted more explicitly: psychologists largely support UMS euthanasia but do not feel adequately prepared, creating a potential gap in care.
Response: We thank the reviewer for this insightful suggestion. We have revised the discussion to explicitly articulate the tension between the predominantly positive attitudes clinical psychologists hold toward UMS euthanasia and their reported gaps in knowledge and practical skills.
Changes: An important tension emerged between the predominantly positive attitudes clinical psychologists expressed toward UMS euthanasia and their reported lack of sufficient knowledge and practical skills to manage such cases effectively. While the mean attitude score indicated strong support for patient self-determination and integration of psychologists within UMS euthanasia care, nearly half of participants (47.1%) reported insufficient knowledge to handle euthanasia requests, and 37.6% indicated lacking necessary practical skills. This gap poses a potential risk to the quality and consistency of care provided to patients requesting UMS euthanasia, underscoring an urgent need for targeted education and training. Furthermore, an overwhelming majority (94.2%) felt that UMS euthanasia was inadequately covered during their formal education or training. Simulation-based learning and specialized curricular modules could help bridge this competence gap, ensuring that positive attitudes translate into clinical readiness to manage the complex ethical, legal, and psychological dimensions of UMS euthanasia. This finding aligns with prior research showing that support for euthanasia does not necessarily equate with professional preparedness, emphasizing the critical importance of capacity building within the clinical psychology workforce (Verhofstadt et al., 2024). Simulation-based education offers valuable experiential learning opportunities, enabling future psychologists to engage with challenging scenarios in a safe, structured environment and to develop confidence and competence under supervision (Demedts et al., 2024; Magerman et al., 2022).
Comment 8: The recommendations for clearer guidelines and simulation-based training are valuable but would be stronger if linked to international examples. For instance, how have training programs in the Netherlands or Canada addressed similar issues?
Response: Thank you for this valuable suggestion. We have now augmented our discussion to link our recommendations for clearer guidelines and simulation-based training to international examples from the Netherlands and Canada.
Changes: Our recommendations for clearer guidelines and enhanced simulation-based training for clinical psychologists on UMS euthanasia align with international efforts in countries like the Netherlands and Canada. In the Netherlands, where UMS euthanasia has been legal since 2002, physicians and nurse practitioners undergo rigorous training aligned with strict legal criteria and due care procedures, supported by institutions such as the Expertisecentrum Euthanasie, which provides ongoing counseling, training, and expertise sharing for clinicians handling complex cases. In Canada, clinician education for (UMS) euthanasia, focuses on understanding legal criteria and safeguards, enhancing skills in capacity assessments specific to mental illness, navigating ethical challenges balancing autonomy and protection, and developing communication abilities for sensitive discussions. This training is delivered through accredited national curricula, such as the Canadian MAID Curriculum developed by the Canadian Association of MAID Assessors and Providers, comprising multiple modules offered via workshops and online sessions to licensed physicians and nurse practitioners nationwide. While Belgium, Canada and the Netherlands have developed nationally accredited, comprehensive training programs for physicians and nurse practitioners involved in (UMS) euthanasia, there are currently no specialized training programs explicitly designed for clinical psychologists.
Comment 9: The call for standardized roles is important but currently presented in broad terms. The authors might suggest specific domains where psychologists could be formally included (e.g., assessment of capacity, family aftercare, supervision of ethical reflection).
Response: Thank you for this insightful comment. We have elaborated on the specific domains in which clinical psychologists could be formally incorporated into the UMS euthanasia care pathway, grounded in both our empirical findings and existing literature.
Changes: A large majority (87.2%) identified themselves as crucial contributors to decision-making processes related to euthanasia requests, particularly emphasizing their role in collaboratively exploring alternatives prior to a final decision—endorsed by 96.3%. Furthermore, 88.0% felt responsible for ensuring that patients are fully informed about the nature and consequences of euthanasia, underscoring their role in patient education and support. Nearly all respondents (98.8%) advocated for a multidisciplinary approach, reflecting the complex ethical and clinical landscape of euthanasia care. Although opinions varied regarding whether psychologists are better positioned than physicians to assess euthanasia eligibility, this diversity underscores the need for clear role definitions. Psychologists were divided on attending euthanasia administration, but a majority (71.1%) considered family aftercare part of their remit. Ethical considerations were prominent, with over three-quarters supporting the right to transfer care in cases of conscientious objection. These findings support the formal inclusion of psychologists in areas such as decisional capacity assessment, collaborative decision-making, patient education, family support, and ethical consultation, consistent with recommendations highlighting psychological expertise in capacity evaluations and ethical deliberations in assisted dying contexts (Galbraith & Dobson, 2000).
Comment 10: Given the sensitivity of the topic, a more explicit reflection on ethical tensions for psychologists themselves (e.g., moral distress, conscientious objection) would enrich the discussion. This is only briefly touched upon in the results.
Response: We appreciate the reviewer’s suggestion to elaborate on the ethical challenges faced by clinical psychologists involved in UMS euthanasia care. We have expanded the discussion to explicitly address the moral distress and ethical tensions experienced by psychologists, drawing on recent empirical and conceptual literature.
Changes: Given the sensitive nature of UMS euthanasia, it is essential to acknowledge the moral distress and ethical tensions experienced by clinical psychologists themselves. Moral distress arises when clinicians perceive conflicts between their professional values—such as preserving life—and the obligation to participate in assisted dying, a phenomenon well documented among healthcare workers involved in euthanasia practices (Pinto et al., 2025). Psychologists may face significant internal conflict, balancing respect for patient autonomy with their own moral and professional commitments. The present finding—that over three-quarters of participants support the right to transfer care due to conscientious objection—reflects an important ethical safeguard protecting clinicians’ moral integrity and mental health. Structured ethical supervision, peer support, and clear institutional policies are vital to mitigating moral distress and sustaining psychological well-being among practitioners engaged in these challenging cases.
Comment 11: The manuscript is generally well written, but some sections could be more concise, particularly the presentation of results where extensive numerical details might be better summarized in text while leaving the full data in tables.
Response: Thank you for this suggestion. We agree with the importance of conciseness in presenting results. However, after initial submission, the editor explicitly requested more detailed reporting of statistical tests and additional analyses to ensure transparency and rigor.
Comment 12: Ensure consistent terminology: sometimes “UMS euthanasia” is used, other times simply “euthanasia.” Clarify at the outset and use consistently.
Response: We have carefully reviewed the manuscript to ensure clarity and consistency in referring to euthanasia in the context of unbearable mental suffering (UMS). At the outset, we clearly define “UMS euthanasia” as euthanasia specifically requested and considered in cases of unbearable mental suffering, distinguishing it from euthanasia related to somatic or terminal conditions.
Changes: Within Belgium, euthanasia may also be granted in cases of unbearable mental suffering (UMS) arising from non-terminal or psychiatric disorders considered both untreatable and hopeless (Gallego et al., 2022). UMS encompasses a wide spectrum of psychological distress, from severe psychiatric illness to existential despair without an identifiable medical diagnosis. It is a deeply subjective and complex phenomenon, relying primarily on patients’ self-reports and clinical interpretation rather than objective biomarkers.
Comment 13: A conceptual diagram summarizing the relationship between attitudes, roles, and competencies could help readers grasp the study framework.
Response: Thank you for your constructive suggestion to include a conceptual diagram summarizing the relationships among attitudes, roles, and competencies of clinical psychologists regarding UMS euthanasia. We agree that such a visual representation enhances reader comprehension of the study framework.
Changes: 3.2. Conceptual Model of Attitudes, Roles, and Competencies of Clinical Psychologists Regarding UMS Euthanasia
Figure 1 presents a conceptual framework illustrating the interrelationships between attitudes, roles, and competencies of clinical psychologists in the context of euthanasia for unbearable mental suffering (UMS). The model visualizes how supportive attitudes—such as endorsement of patient autonomy and underlying ethical beliefs—influence psychologists’ perceptions and enactment of their professional roles within multidisciplinary euthanasia care teams. Concurrently, knowledge, practical skills, and training constitute critical competencies that enhance role acceptance and effective participation in the euthanasia process. Ethical tensions, specifically the balance between respecting patient autonomy and protecting vulnerable individuals, mediate the dynamics between attitudes and role perceptions.
Comment 14: The reference list is up to date and relevant, but a few important philosophical and ethical sources (e.g., Beauchamp & Childress on biomedical ethics, discussions of autonomy in psychiatry, etc.) could strengthen the theoretical grounding.
Response: Thank you for this valuable recommendation. We have integrated key sources such as Beauchamp and Childress’s Principles of Biomedical Ethics to more explicitly frame the ethical tensions surrounding autonomy, beneficence, and protection in UMS euthanasia.

Reviewer 3 Report
Comments and Suggestions for Authors
Dear authors:
Congrats on your work. Here are my comments/suggestions:
Abstract: Please structure the abstract as suggested by the MDPI group guidelines. It makes the abstract more understandable and easier to read.
Introduction: The end of your introduction should support your conclusion; in other words, it should highlight the main advantages or findings of your study. The introduction should conclude with the objective of your research, ensuring consistency with what has already been described in the abstract.
Methods: The methodology could be more detailed and descriptive, for example, by specifying what types of statistical measures were used in the data analysis.
Results and discussion: well written. Please add limitations of your study at the end of discussion.
Conclusion: See my previous comments regarding introduction, please.
Author Response
Comment 1: Please structure the abstract as suggested by the MDPI group guidelines. It makes the abstract more understandable and easier to read.
Answer: Thank you for bringing this to our attention. We have reviewed the specific guidelines for the 'European Journal of Investigation in Health, Psychology and Education' and believe that the abstract has been structured in accordance with their requirements. However, these differ from those of other MDPI journals, such as 'Healthcare'.
Comment 2: The end of your introduction should support your conclusion; in other words, it should highlight the main advantages or findings of your study. The introduction should conclude with the objective of your research, ensuring consistency with what has already been described in the abstract.
Answer: Thank you for this valuable comment. We have revised the conclusion of the introduction to clearly state the objective of the research, including the specific research question.
Changes:
While the euthanasia debate is global, Belgium provides a particularly instructive legal and healthcare context, featuring a mature regulatory framework and established multidisciplinary procedures. Conducting this study in Belgium therefore offers a valuable opportunity to explore psychologists’ professional views within one of the world’s most developed systems of (UMS) euthanasia governance. Findings are expected to enrich international understanding despite contextual differences across countries. Guided by these considerations, the present study addresses the following question: What are the attitudes, roles, and competencies of clinical psychologists regarding UMS euthanasia?.
Comment 3: The methodology could be more detailed and descriptive, for example, by specifying what types of statistical measures were used in the data analysis.
Answer: Thank you for your constructive feedback. In response, we have revised the Methodology section to include more detailed descriptions of the statistical measures and procedures used in the data analysis. Specifically, we added information about the calculation of descriptive statistics and the use of one-way ANOVA and independent samples t-tests to examine group differences. We also clarified that effect sizes (partial eta squared and Cohen’s d) were calculated to quantify the magnitude of observed differences, with statistical significance set at p < 0.05.
Changes:
Participants first answered demographic questions (age, gender, healthcare echelon, palliative care team membership, specialized psychiatric work, and prior euthanasia involvement) before completing the main survey. The UMS-EAS-NL questionnaire comprised 38 items adapted from prior research on UMS euthanasia attitudes in nursing students (Demedts et al., 2023a; Demedts et al., 2023b) and derived from the validated Euthanasia Attitude Scale (Tordella & Neutens, 1979; Rogers, 1996). The attitude section included four thematic domains: Ethical considerations, Practical considerations, Treasuring life, and Naturalistic beliefs (Chong & Fok, 2005). Items were rated on a 5-point Likert scale (1 = strongly disagree to 5 = strongly agree), with negatively worded items reverse-coded. Higher scores indicated greater acceptance of UMS euthanasia.
The scale demonstrated good internal consistency (Cronbach’s alpha = 0.812; McDonald’s omega = 0.838) and construct validity in previous studies (Demedts et al., 2023a). It also differentiated attitudes by prior euthanasia involvement in the current sample. The second section assessed perceived roles and competencies, developed through literature review and expert consensus with adaptations for clinical psychologists, also rated on a 5-point Likert scale (Demedts et al., 2023b).
Descriptive statistics (means, standard deviations, frequencies) summarized participant characteristics and survey responses. Attitude analyses involved comparing each thematic domain’s scores against a neutral midpoint (“neither agree nor disagree”) to determine overall positive or negative stances. Group comparisons were conducted using one-way ANOVA tests for demographic variables such as age groups and healthcare echelon, and independent samples t-tests to assess differences based on gender, palliative care team membership, specialized psychiatric work, and prior euthanasia involvement. Effect sizes (partial eta squared for ANOVAs and Cohen’s d for t-tests) were calculated to quantify the magnitude of differences. Statistical significance was accepted at p < 0.05. All data were collected anonymously using Qualtrics and analyzed with IBM SPSS Statistics version 29.0.
Comment 4: Results and discussion: well written. Please add limitations of your study at the end of discussion.
Answer: Thank you for this insightful suggestion. In response, we have added a dedicated limitations section at the end of the discussion. This section addresses key considerations including the cross-sectional design, reliance on self-reported data, potential sample bias due to recruitment through professional associations in Flanders, and the adaptation of the survey instrument from nursing research. We also acknowledge the omission of potentially influential unmeasured variables such as personal values and cultural factors.
Changes:
This study has several limitations that should be considered when interpreting the findings. First, the cross-sectional design provides only a snapshot of attitudes and competencies at a single point in time, limiting insight into how these may evolve. Second, the exclusive use of self-report questionnaires may introduce response biases, particularly social desirability bias, which could inflate positive attitudes or perceived competencies. Additionally, discrepancies may exist between participants’ self-reported attitudes and their actual clinical practices, as intentions do not always translate into behavior. Third, the sampling strategy relied heavily on professional associations within Flanders, potentially introducing selection bias. Members of these associations may be more professionally engaged or possess greater interest or expertise in euthanasia, which could limit the representativeness of our sample and constrain the generalizability of findings to the wider population of clinical psychologists in the region. Our sample of 242 actively practicing clinical psychologists represents only a small portion of the estimated total registered population. Although this constitutes a significant proportion, the response rate could not be precisely calculated due to reliance on organizational sharing and voluntary participation. Fourth, important factors such as personal values, cultural or religious beliefs, and previous experiences with euthanasia were not measured but might have influenced responses. Previous research has demonstrated that religiosity and cultural background strongly shape healthcare professionals’ attitudes toward euthanasia (Dopelt et al., 2021). Religious beliefs often provide fundamental ethical frameworks influencing acceptance or opposition to euthanasia, with more religious individuals tending to hold more negative attitudes. These cultural and religious dimensions are critical for fully understanding professional perspectives and should be incorporated in future studies. Fifth, although the survey instrument was adapted from measures previously validated in nursing student populations—showing good psychometric properties—it was not subjected to a separate pilot study or additional reliability testing within this professional group. Future validation efforts should focus on establishing this instrument’s reliability and validity specifically among psychologists. Finally, future research employing longitudinal, mixed-method designs and larger, more diverse samples would deepen understanding of the complex and sensitive issues surrounding psychologists’ roles in UMS euthanasia.
Comment 5: Conclusion: See my previous comments regarding introduction, please.
Answer: Thank you for your constructive comment. We have revised the conclusion to integrate the key findings while acknowledging the identified limitations of the study.
Changes:
This study provides novel empirical insights into the attitudes, roles, and competencies of clinical psychologists in Flanders regarding UMS euthanasia. The findings reveal broad professional endorsement of patient self-determination and strong support for integrating clinical psychologists within multidisciplinary euthanasia care teams. Nevertheless, significant gaps in knowledge, practical skills, and educational preparation were identified. Addressing these deficiencies through clearer role delineations, the development of standardized national guidelines, and enhanced training programs—including simulation-based learning—could enhance both professional competency and the quality of care offered to patients and their families. Although findings are based on a heterogeneous sample of psychologists from diverse care settings, their generalizability is limited by the study’s design and sampling framework. Therefore, further research using larger, more representative samples and qualitative methods is warranted to comprehensively clarify clinical psychologists’ roles in UMS euthanasia and to inform policy and practice at national and international levels.

Round 2
Reviewer 1 Report
Comments and Suggestions for Authors
The authors addressed all the comments.
Reviewer 2 Report
Comments and Suggestions for Authors
The manuscript has undergone substantial revisions in response to the given feedback. The authors successfully transformed the content into a more critically positioned, methodologically transparent, and conceptually grounded scientific contribution.
The revisions demonstrate a significant increase in the article's rigor, notably in the following domains: (1) Theoretical and Ethical Framing: The introduction is now anchored in the foundational bioethical debate of autonomy versus protection in the context of euthanasia for Unbearable Mental Suffering (UMS). This provides a more robust framework, and the research gap concerning clinical psychologists (versus physicians/nurses) is clearly established, strengthening the study's rationale; (2) Addressing the Competency Gap (Core Finding): The discussion is effectively centered on the critical tension between psychologists' strong supportive attitudes toward UMS euthanasia and their self-reported lack of sufficient knowledge and practical skills. This compelling finding creates an urgent call for action and forms the primary takeaway of the paper; (3) Actionable and Grounded Recommendations: Recommendations for the formal role of psychologists are now more specific (e.g., capacity assessment, family aftercare, ethical consultation). The call for simulation-based training is directly linked to the competency deficit and supported by international precedents from countries like the Netherlands and Canada; (4) Methodological Transparency: The authors demonstrated strong scientific integrity by explicitly acknowledging limitations related to selection bias (professional association sampling), social desirability bias in self-report measures, and the intention-behavior gap inherent to cross-sectional designs; (5) Ethical Depth: The manuscript now includes a necessary reflection on the ethical burden for practitioners, including moral distress and the importance of conscientious objection as an ethical safeguard, ensuring a nuanced perspective on this sensitive topic.